# Pectin Nanoparticle-Loaded Soft Coral *Nephthea* sp. Extract as *In Situ* Gel Enhances Chronic Wound Healing: *In Vitro*, *In Vivo*, and *In Silico* Studies

**DOI:** 10.3390/ph16070957

**Published:** 2023-07-03

**Authors:** Nevine H. Hassan, Seham S. El-Hawary, Mahmoud Emam, Mohamed A. Rabeh, Mohamed A. Tantawy, Mohamed Seif, Radwa M. A. Abd-Elal, Gerhard Bringmann, Usama Ramadan Abdelmohsen, Nabil M. Selim

**Affiliations:** 1Pharmacognosy Department, Faculty of Pharmacy, Modern University for Technology and Information, Cairo 11571, Egypt; nevine.hossam@pharm.mti.edu.eg; 2Pharmacognosy Department, Faculty of Pharmacy, Cairo University, Giza 11562, Egypt; seham.elhawary@pharma.cu.edu.eg; 3Phytochemistry and Plant Systematics Department, National Research Centre, Dokki, Cairo 12622, Egypt; mahmoudemamhegazy2020@gmail.com; 4Pharmacognosy Department, College of Pharmacy, King Khalid University, Abha 62514, Saudi Arabia; mrabeh@kku.edu.sa; 5Hormones Department, Medical Research and Clinical Studies Institute, National Research Centre, Dokki, Cairo 12622, Egypt; mohamed_tantawy@daad-alumni.de; 6Stem Cells Lab Center of Excellence for Advanced Sciences, National Research Centre, Dokki, Cairo 12622, Egypt; 7Center of Orthopaedics Research, and Translation Science (CORTS), Department of Orthopaedics and Rehabilitation, The Pennsylvania State University College of Medicine, State College, PA 16801, USA; 8Toxicology and Food Contaminants Department, Food Industries and Nutrition Research Institute, National Research Centre, Giza 12622, Egypt; seif.eg@gmail.com; 9Pharmaceutics and Drug Manufacturing Department, Faculty of Pharmacy, Modern University for Technology and Information, Cairo 11571, Egypt; 10Institute of Organic Chemistry, University of Würzburg, Am Hubland, 97074 Würzburg, Germany; bringman@chemie.uni-wuerzburg.de; 11Pharmacognosy Department, Faculty of Pharmacy, Minia University, Minia 61519, Egypt; 12Pharmacognosy Department, Faculty of Pharmacy, Deraya University, New Minia 61111, Egypt

**Keywords:** ADME, *in situ* gel, molecular docking, *Nephthea* sp., pectin nanoparticles, wound healing

## Abstract

This study shed light for the first time on the *in vivo* diabetic wound healing potential activity of natural marine soft coral polymeric nanoparticle *in situ* gel using an excision wound model. A *Nephthea* sp. methanol–methylene chloride extract loaded with pectin nanoparticles (LPNs) was created. For the preparation of *in situ* gel, ion-gelation techniques, the entrapment efficiency, the particle size, the polydispersity index, the zeta potential, the *in-vitro* drug release, and a transmission electron microscope were used and the best formula was selected. Using (UPLC-Q/TOF-MS), 27 secondary metabolites responsible for extract biological activity were identified. Isolation and identification of arachidic acid, oleic acid, nervonic acid, and bis-(2-ethylhexyl)-phthalate (DEHP) of *Nephthea* sp. was firstly reported here using NMR and mass spectral analyses. Moreover, LPN *in situ* gel has the best effects on regulating the proinflammatory cytokines (NF-κB, TNF-α, IL-6, and IL-1β) that were detected on days 7 and 15. The results were confirmed with an *in vitro* enzymatic inhibitory effect of the extract against glycogen synthase kinase (GSK-3) and matrix metalloproteinase-1 (MMP-1), with IC_50_ values of 0.178 ± 0.009 and 0.258 ± 0.011 µg/mL, respectively. The molecular docking study showed a free binding energy of −9.6 kcal/mol for chabrolosteroid E, with the highest binding affinity for the enzyme (GSK-3), while isogosterone B had −7.8 kcal/mol for the enzyme (MMP-1). A pharmacokinetics study for chabrolohydroxybenzoquinone F and isogosterone B was performed, and it predicted the mode of action of wound healing activity.

## 1. Introduction

Skin wounds are considered as a global health issue due to their ability to kill more than 5 million people each year and contribute significantly to disease burden mainly to people of a low socioeconomic status [1]. Wounds are classified as physical, thermal, or chemical injuries that create an opening in the integrity of the skin or change the anatomical integrity of living tissues [2]. The plurality of chronic wounds are ulcers produced by diabetes mellitus, pressure, ischemia, and excessive obesity [3]. Wound healing medicines must be biocompatible, non-toxic, hypoallergenic, able to maintain a moist environment, protect the wound from germs, and encapsulate wound exudates [4]; this established the urgent need for the development of innovative safe therapies in this field [5].

Nanoscience is one of the most recent sciences to pique the interest of scientists [6]. This is a brand new field that involves the creation and application of nanoscale-size materials to a variety of applications, and they are especially hygienic, non-toxic, and ecologically friendly [7]. Natural resources have the potential to be used to create nanoparticles that are bioactive and compatible with biological systems but this potential has not been completely examined [8,9].

Pectin is a naturally anionic water-soluble polysaccharide polymer, derived from the cell walls of certain citrus plants [10]. Due to its cross-linked properties, non-toxicity, bio-compatibility, mucoadhesion properties, water absorption capacity, moisture retention, and anti-inflammatory activity [11], pectin is an attractive choice to form polymeric pectin nanoparticles to encapsulate both hydrophilic and lipophilic compounds [12].

Smart polymers have been used to create stimuli-responsive *in situ* gels; these polymers tend to change their consistency in response to physiological changes. The *in situ* gels created with these polymers initially take the form of a solution and after being administered to the skin, go through a phase transition (from sol to gel) in response to heat stimulation, pH changes, or ionic concentration [13].

The genus *Nephthea* has been discovered to produce secondary metabolites like sesquiterpenes, fatty acids, terpenes, and steroids [14]. Additionally, a review on the scientific literature on the biological efficacy of the octocoral *Nephthea* sp. revealed that it can act as an inhibitor of dermatophytes [15] as well as an antiviral [16], antibiofilm [17], anti-COVID-19 [18], and gastroprotective agent [19].

To the best of our knowledge, the wound healing potential activity of the soft coral *Nephthea* sp. has not been previously evaluated. Therefore, the novelty of our study is the investigation of the chronic wound healing stimulation activity of the pharmaceutical preparation of polymer nanoparticle-loaded *Nephthea* sp., which was obtained from the Red Sea region. The pectin nanoparticle-loaded *Nephthea* sp. extract prepared with an ion-gelation technique at 0.5% *w/v* of pectin with CaCl_2_ as a cross-linker in a different ratio was elucidated with *in vitro* studies, including the entrapment efficiency, particle size, polydispersity index, zeta potential, *in vitro* release study, and morphological study with a transmission electron microscope, to select the optimized formulation, and it was incorporated as an *in situ* gel dosage form for transdermal drug delivery in diabetic-wound rats. Additionally, the metabolic profile of the *Nephthea* sp. extract was examined using the UPLC-Q/TOF-MS technique, then the major identified components were isolated. Additionally, NF-κB and pro-inflammatory cytokines (TNF-α, IL-6, and IL-1β) were measured for the different treatments in the middle and at the end of the experiment. Furthermore, the inhibitory action of the extract against GSK-3 and MMP-1 enzymes was tested to confirm the mechanism of action toward wound healing activity. Finally, an *in silico* study of the identified metabolites against GSK-3 and MMP-1 enzymes was assembled, followed by pharmacokinetics and an ADME profile of the high-binding components (Figure 1).

## 2. Results and Discussion

### 2.1. Preparation and Characterization of SCN-LPN

The ion-gelation method was successful in loading SCN within the formed pectin nanoparticles with different ratios of CaCl_2_ as a cross-linker. This might be related to the positively charged CaCl_2_, which was successfully cross-linked with the negatively charged pectin molecules by an intermolecular cross-link, which is formed by electrostatic complexation interaction [20].

The % E.E. indicates how the formed particles can load sufficient quantities of the drug. As shown in Table 1, all the formed SCN-LPN formulations had an E.E. value higher than 90%, and increasing the pectin-to-CaCl_2_ ratio from 1:1 to 1:3 had a non-significant difference on the E.E. value (*p* > 0.05). This could be related to the hydrophobic nature of SCN, which contains sesquiterpenes, diterpenes, and sterols, so it can easily entrap into polymeric nanoparticles during a sudden electron complexation between negatively charged pectin molecules and CaCl_2_. Thereby, a limited drug leakage from the formed polymeric nanoparticles occurred.

The formed SCN-LPN had acceptable PS, ranging between 241.5 ± 17.5 and 288.2 ± 25.66 nm with low PDI values ranging from 0.290 ± 0.22 to 0.523 ± 0.12, which indicated the homogeneity of the formed SCN-LPN formulations. In addition to this, it was observed that raising the pectin-to-CaCl_2_ ratios from 1:1 to 1:2 was associated with a significant increase in the particle size of the formed SCN-LPN (*p* < 0.05). This could be related to an enhanced number of CaCl_2_ molecules that interacted electrostatically with the pectin chains, resulting in the formation of a large SCN-LPN size [21]. However, with any further increase in the pectin-to-CaCl_2_ ratio above 1:2, a significant decrease in PS was observed. There was a direct relation between the amount of SCN loaded within the formed nanoparticles and their particle size, as shown in Table 1.

The ZP value is a crucial parameter of nanoparticles, in addition to PS, since it can show the stability of the nanoparticle formulations [22,23]. Higher absolute ZP values can strengthen the stability of the system related to a strong electrostatic repulsion between the particles that also prevent the agglomeration between them, or it could be related to the ionization of carboxyl groups of pectin [21]. All the SCN-LPN formulations had a negative charge of ZP ranging between −5.34 ± 1.20 and −16.6 ± 2.4 mV due to the ionization of the carboxyl group of pectin. Increasing the pectin-to-CaCl_2_ ratio was accomplished by decreasing the ZP value of the prepared SCN-LPN (*p* < 0.05). This could be explained with a strong neutralization of the negative charges of the free carboxyl groups of pectin with divalently charged CaCl_2_ as a cross-linker [21,24]. From the previous results, the selected formulation for further studying was SCN-LPN2 with a % E.E. of 98.55 ± 2.20, PS of 288.2 ± 25.66, PDI of 0.381 ± 0.12, and ZP of −16.6 ± 2.4, as shown in Figure 1.

#### 2.1.1. HRTEM Characterization of the Selected SCN-LPN

The morphology of the selected formula (SCN-LPN2) was evaluated with an HRTEM analysis and is depicted in Figure 2. It showed that the prepared nanoparticles were uniform and spherical in shape and no agglomeration was observed. The mean particle size was obtained with the HRTEM analysis of nanosized particles; this followed the results obtained with Zeta-sizer equipment. In addition, selected area electron diffraction (SAED) confirmed that the prepared formulation had an amorphous property with no SCN crystals being observed.

#### 2.1.2. UV-Visible and FTIR Characterization of the Selected SCN-LPN Formulation

The chemical structure of the selected SCN-LPN formulation was evaluated with UV-Vis and FTIR spectroscopy. The formation of SCN-LPN2 was monitored by measuring the UV-Vis spectrum of the reaction medium in the wavelength range from 200 to 600 nm. Figure 3 illustrates the UV-Vis spectra of the *Nephthea* sp. extract, showing an absorption maximum peak at 415 nm, while the unloaded formula exhibited a λ_max_ band at 350 nm. Upon the interaction between both the mixtures “unloaded and the extract” being performed, the observed bands disappeared, indicating that the interaction between the two mixtures was achieved and the “loaded” SCN-LPN2 formula was prepared.

Using FTIR spectroscopy, the chemical structure of the selected loaded formulation was investigated and compared with the unloaded formulation, extract, CaCl_2_, and pectin structures; see Figure 4. The peaks at 3351, 1716, and 1644 cm^−1^ in the pectin, extract, and unloaded samples were observed, indicating the presence of OH and C=O functional groups, while after adding the mixture of the extract and unloaded formula, these observed peaks were shifted and shortened, which indicated that the interaction was taking place. These spectroscopic findings supported the creation of the SCN-LPN.

#### 2.1.3. Characterization of SCN-LPN-ISG

SCN-LPN2-ISG appeared as a homogeneous, yellowish clear system related to the color of SCN at a suitable pH value (6.5 ± 0.5) for skin application without any irritation. The drug content value was 99.43 ± 1.1, indicating the uniformity of distribution during the preparation of *in situ* gel. The *sol–gel* transition temperature of the prepared SCN-LPN2-ISG was 32.20 °C ± 0.65, ideal for skin application, and this could be due to a suitable combination ratio between PF^®^127 and PF^®^68, because PF^®^68 could decrease the total polymer content and improve gelling properties of PF^®^127.

The viscosities of the SCN-LPN2-ISG freshly prepared before gelling (25 °C) at 1 rpm and 10 rpm were found to be 82.55 ± 11.70 and 40.66 ± 6.88, respectively. The viscosities were raised to 19,234.22 ± 210.24 and 2805.55 ± 177.63, respectively, as a response to the temperature (37 °C), meaning that the viscosity of the prepared SCN-LPN2-ISG was drastically increased when raising the temperature from 25 °C to 37 °C, where gelling will occur, owing to the thermo-sensitive properties of the prepared ISG.

The cumulative amount of SCN (%) released from the SCN suspension, selected SCN-LPN2, and SCN-LPN2-ISG at different time intervals is graphically illustrated in Figure 5. Almost 90% of the SCN amount was released from the SCN suspension in the first 2 h. However, the selected SCN-LPN2 and SCN-LPN2-ISG released around 30% of the loaded drug in the first 2 h and succeeded in retarding the remaining loaded drug for 8 h in comparison with the SCN suspension. This could be related to the nanoparticle matrix, which acts as a barrier to drug release toward the dissolution medium [25].

### 2.2. Metabolomic Profiling Study

The UPLC-Q/TOF-MS profile of the *Nephthea* sp. extract is shown in Figure 6 and Appendix A, and Table 1. Table 1 provides the retention times, identities, observed molecular weights, and ionization modes for the recognized metabolites. Using macros, MZmine-based methods, and internet databases (databases DNP and MarinLit), 27 metabolites were identified, the majority of which were compared with previously reported published data from the Nephtheidae family and classified as described in the following reference: [16].

Sterols possessing the mass ions of *m/z* 429.336, 505.315, 395.653, 441.299, 501.284, 443.314, 427.319, 429.336, 399.420, 517.351, 413.274, 471.875, and 503.254 were in agreement with the molecular formulas C_28_H_44_O_3_, C_29_H_46_O_7_, C_28_H_42_O, C_28_H_40_O_4_, C_29_H_42_O_7_, C_28_H_42_O_4_, C_28_H_42_O_3_, C_28_H_44_O_3_, C_27_H_42_O_2_, C_31_H_48_O_6_, C_30_H_48_O_4_, and C_31_H_50_O_5_. They were dereplicated as erectasteroid H (**5**), pregn-20-ene-3,19-diol, (3β,5α) form, 3-*O*-(3-*O*-acetyl-α-L-fucopyranoside) (**6**), ergosta-1,4,24(28)-trien-3-one (**7**), chabrolosteroid H (**8**), isogosterone B (**9**), chabrolosteroid G (**11**), chabrolosteroid E (**14**), chabrolosteroid C (**15**), cholest-1-ene-3,22-dione (**16**), nanjiol B (**21**), nephalsterol C (**25**), and nebrosteroid B (**26**), respectively.

Diterpenes with expected chemical formulas of C_20_H_28_O_2_ and mass ions of *m/z* 221.376, 439.283, 271.346, 427.284, and 379.896 were dereplicated as nephthenol (**1**), chabrolobenzoquinone B (**4**), cembrene A (**10**), chabrolohydroxybenzoquinone F (**13**), and pacificin G (**17**), respectively.

Moreover, further fatty acid, providing the detected mass ions of *m/z* 281.75449, 355.320, 311.054, 281.245, and 365.905, corresponding to the chemical formulas C_18_H_34_O_2_, C_22_H_44_O_3_, C_20_H_40_O_2_, C_18_H_32_O_2_, and C_24_H_46_O_2_, respectively, components were dereplicated as oleic acid (**18**), 2-hydroxydocosanoic acid (**19**), arachidic acid (**20**), linoleic acid (**23**), and nervonic acid (**24**), respectively.

The observed mass ions of *m/z* 249.149 and 237.542 and the chemical formulas C_15_H_20_O_3_ and C_15_H_24_O_2_ evidenced the presence of the sesquiterpenes armatin F (**3**) and orientalol C (**22**), which have been identified earlier in *Nephthea* sp. [26].

A deprotonated molecule [M-H]^−^, with an expected chemical formula of C_36_H_73_NO_4_ and a mass ion of *m/z* 582.763, was identified as the ceramide nephtixamide B (**27**), while ketochabrolic acid (**2**), which can be classified as a terpenoid-related carboxylic acid, appeared as a protonated molecule [M+H]^+^ with a mass ion of *m/z* 293.227 and with a chemical formula of C_18_H_28_O_3_. Also, the protonated molecule [M + H]^+^ bis(2-ethylhexyl)phthalate (**12**), with the chemical formula C_24_H_38_O_4_, was detected with a mass ion of *m/z* 392.539.

#### Isolation of the Major Metabolites

Compound **12** was isolated and was identified as bis(2-ethylhexyl)phthalate (DEHP) with ESI-MS as well as ^1^H and ^13^C NMR analysis data [27]. The ESI-MS data presented a molecular ion peak at *m/z* 391.42 and a base peak at 148.84 (calc. for C_24_H_38_O_4_, 267.0657). Appendix A shows the ^1^H and ^13^C NMR spectral data, which agree with findings of Elhagali et al., 2019 [28].

Appendix A show the ESI-MS spectra of compounds **18** and **20**, which were recognized as oleic acid and arachidic acid, respectively, by comparing the base peak with published data [29,30].

Based on characteristic signals in the ^1^H and ^13^C NMR spectra data and using comparison with the published data, compound **24** was identified as nervonic acid [31,32]. These data are displayed in Appendix A. This study is the first report on the isolation of these four compounds from the genus *Nephthea*.

**Table 1 pharmaceuticals-16-00957-t001:** Tentatively identified secondary metabolites identified in *Nephthea* sp. extract (using UPLC–Q/TOF–MS).

No.	Identified Metabolites	Molecular Formula	Rt. (min)	Ionization Mode	*m/z*	Molecular Weight	∆Mass (ppm)	Chemical Class	References
**1**	Nephthenol	C_15_H_26_O	5.56	Negative	221.376520	222.198365	−2.3185972	Cembrane diterpene	[33]
**2**	Ketochabrolic acid	C_18_H_28_O_3_	5.75	Positive	293.227486	292.203803	−1.9893311	Terpenoid-related carboxylic acids	[34]
**3**	Armatin F	C_15_H_20_O_3_	6.48	Positive	249.149275	248.141930	2.7641515	Sesquiterpene	[26]
**4**	Chabrolobenzoquinone B	C_28_H_38_O_4_	7.79	Positive	439.283800	438.276500	−1.1620504	Diterpene	[34]
**5**	Erectasteroid H	C_28_H_44_O_3_	8.31	Positive	429.336642	428.328765	−0.6532361	Steroid	[26]
**6**	Pregn-20-ene-3,19-diol; (3β,5α)-form, 3-*O*-(3-*O*-acetyl-α-L-fucopyranoside)	C_29_H_46_O_7_	9.11	Negative	505.315812	506.323088	−2.5011635	Steroid	[14]
**7**	Ergosta-1,4,24(28)-trien-3-one	C_28_H_42_O	9.26	Positive	395.653219	394.323565	−1.9664193	Steroid	[35]
**8**	Chabrolosteroid H	C_28_H_40_O_4_	9.43	Positive	441.299122	440.291794	−1.9664193	Steroid	[34], [36]
**9**	Isogosterone B	C_29_H_42_O_7_	9.87	Negative	501.284708	502.291984	−2.1308277		
**10**	Cembrene A	C_20_H_32_	10.35	Negative	271.346700	272.250976	−2.8617931	Cembrane diterpene	[37]
**11**	Chabrolosteroid G	C_28_H_42_O_4_	10.41	Positive	443.314955	442.307657	−1.4758935	Steroid	[34]
**12**	Bis(2-ethylhexyl)phthalate *	C_24_H_38_O_4_	10.74	Positive	391.539732	390.426789	−2.9895325	Phthalates	[27]
**13**	Chabrolohydroxybenzoquinone F	C_27_H_40_O_4_	11.45	Negative	427.284428	428.291704	−2.2307176	Diterpene	[34]
**14**	Chabrolosteroid E	C_28_H_42_O_3_	11.56	Positive	427.319666	426.312288	−2.5948047	Steroid	[34]
**15**	Chabrolosteroid C	C_28_H_44_O_3_	12.04	Positive	429.336311	428.329034	−0.0238134		
**16**	Cholest-1-ene-3,22-dione	C_27_H_42_O_2_	12.35	Positive	399.420384	398.318485	0.4476480	Steroid	[14]
**17**	Pacificin G	C_22_H_36_O_5_	12.65	Negative	379.896542	380.256275	−1.3900752	Diterpene	[26]
**18**	Oleic acid *	C_18_H_34_O2	12.95	Negative	281.754231	282.586432	−2.0964387	Fatty acid	[15]
**19**	2-Hydroxydocosanoic acid	C_22_H_44_O_3_	13.26	Negative	355.320800	356.328077	−2.7163095	Fatty acid	[38]
**20**	Arachidic acid *	C_20_H_40_O_2_	13.58	Negative	311.054217	312.502653	−1.9700365	Fatty acid	[15]
**21**	Nanjiol B	C_31_H_48_O_6_	13.42	Positive	517.351338	516.343721	−2.6503592	Steroid	[26]
**22**	Orientalol C	C_15_H_24_O_2_	14.43	Positive	237.542987	236.177638	−2.5296721	Sesquiterpene	[39]
**23**	Linoleic acid	C_18_H_32_O_2_	14.86	Positive	281.245678	280.240232	1.5898976	Fatty acid	[15]
**24**	Nervonic acid *	C_24_H_46_O_2_	14.32	Negative	365.905392	366.219853	−2.9065342	Fatty acid	[15]
**25**	Nephalsterol C	C_30_H_48_O_4_	15.39	Negative	471.875324	472.355263	−2.9850569	Steroid	[40]
**26**	Nebrosteroid B	C_31_H_50_O_5_	15.53	Positive	503.254835	502.365825	−0.2345662	Steroid	[14]
**27**	Nephtixamide B	C_36_H_73_NO_4_	16.46	Negative	582.763198	583.553959	−1.9956883	Ceramide	[18]

* Compounds isolated in this study.

### 2.3. Wound Healing Activity

The diabetic diagnostic was confirmed when the glucose level exceeded 300 mg/dL. During the experiments, the glucose level was monitored daily. The insulin was injected only when the level exceeded 500 mg/dL, in order to protect the rats against weakness and sudden death.

The current findings indicate that in all experimental groups, wound closure rates increased in a time-dependent manner. The wound closure percentages were about 18 to 20% in each group on day 3 after injury, with no discernible distinctions between the groups, with the lowest being in the untreated group and the highest in the treated ones. However, the wound closure in the SCN-LPN2-ISG-treated group reached 50% on day 6 after treatment, which was significantly higher than the corresponding untreated group; see Table 2 and Figure 7 and Figure 8.

Additionally, the SCN-LPN2-ISG-treated group also showed high wound closure percentages compared to those of the *Nephthea* sp. extract and MEBO-treated group. The centripetal flow of the edges of a full-thickness wound to aid in wound tissue closure is referred to as wound closure. Figure 8 discloses the percentage of wound healing of the SCN-LPN2-ISG-treated group (80%) was significantly greater than that of the untreated group (50%) on the 13th day after injury. On day 15, the wounds in the treated groups were perfectly cured and the wound closure was 95% in the skin of rats that were treated with SCN-LPN2-ISG and 90% in the MEBO-treated group.

#### 2.3.1. Histopathological Examination of Skin Tissue

The control healthy group (non-wounded) had no histopathological alteration and a normal histological structure of the epidermis with the underlying dermis with hair follicles and sebaceous glands followed by musculature and subcutaneous tissue (Figure 9A). On the other hand, Figure 9C shows the MEBO-treated group with a few underlying inflammatory cells’ infiltration and granulation tissue formation as well as a loss of the hair follicle, and inflammatory cell infiltration was detected in subcutaneous tissue (Figure 9C). A mild focal acanthosis was detected in the epidermis of the SCN-LPN2-ISG group, while the underlying dermis showed formation of granulated tissue and a loss of hair follicles and sebaceous glands in association with a few infiltrations of inflammatory cells into the subcutaneous tissue (Figure 9E).

Moreover, the *Nephthea* sp. extract exposed a slight infiltration of inflammatory cells in the subcutaneous tissue and the deep dermis had granulation tissue formation with a loss of hair follicles (Figure 9D), compared with the untreated group, which showed focal necrosis in the epidermal and dermal layers with massive inflammatory cell infiltration (Figure 9B). According to these histological studies, SCN-LPN2-ISG and MEBO have a potent chronic wound healing effect.

#### 2.3.2. Inflammation Markers

Inflammation is an evolutionarily settled process that allows for the defense of an organism against damage to its tissues and organs. The progression of wound inflammation is crucial for the optimal completion of hemostasis, as well as the detection and elimination of pathogenic microorganisms, the removal of damaged tissues, and wound cleaning [41]. These steps progress via the participation of leukocytes, which migrate from the bloodstream to the site of injury. The migration is followed by the formation and release of pro-inflammatory cytokines and phagocytosis [42].

The current findings, as illustrated in Figure 10, show the inflammation response during the wound healing process under different treatments on days 7 and 15. The level of NF-κB was significantly decreased in groups designedly under treatment with the MEBO, extract, or SCN-LPN2-ISG compared to the untreated wounded group. Moreover, the TNF-α in the MEBO-treated group showed a non-significant difference compared with untreated wounded animals, while treatment with the extract and SCN-LPN2-ISG led to a significant decrease compared with untreated wounded animals on day 7; the same trend was found on day 15, with a noteworthy decrease when compared.

In the same trend, the treatment with the SCN-LPN2-ISG and extract caused a significant decrease in the IL-6 level when compared with MEBO-treated and untreated-wound groups; the same trend was noted on day 14. Furthermore, the different treatments did not cause any substantial different effects on the serum level of IL-1β on day 7, and the same trend was recorded on day 15, with a significant decrease (*p* < 0.05).

Proinflammatory cytokines are among the first factors to be produced in response to skin wounds. They regulate the functions of immune cells in epithelialization. Proinflammatory cytokines, including TNF-α, are a central factor in the macrophage-promoted hair follicle telogen–anagen transition, and contribute to hair follicle neogenesis in skin wound healing [43]. Proinflammatory cytokines (TNF-α, IL-6, and IL-1β) primarily play a pivotal role in acute wound healing by promoting the proliferation and antimicrobial peptide production of keratinocytes. An overproduction of proinflammatory cytokines, however, may lead to prolonged inflammation and wound healing. Therefore, blocking excessive proinflammatory cytokines exerts a therapeutic effect in chronic wound healing, which was observed to occur in the extract- and SCN-LPN2-ISG-treated groups; the effects may be attributed to anti-inflammatory active components that were found in the extract of the soft coral *Nephthea* sp. [44]. Inflammation is a vital action occurring during wound healing, and it is necessary to remove the apoptotic cells from the damaged area [45]. According to the observation in Figure 7 and Figure 10, the relation between the wound statuses was parallel with the level of inflammation markers. All tested inflammation protein was decreased at day 15 compared to them on day 7. The decrease in the inflammation markers was in parallel with the degree of the healing of wounds at the 15th day compared to them at the 7th day.

### 2.4. In Vitro Inhibition Assay of GSK-3 and MMP-1 Enzymes

Rapid hemostasis, appropriate inflammation, mesenchymal cell differentiation, proliferation and migration to the wound site, suitable angiogenesis, prompt re-epithelialization, and proper collagen synthesis, cross-linking, and alignment are all pivotal requirements for optimal wound healing in healthy persons [3]. The strength of the rebuilding tissue is also the inhibition of glycogen synthase kinase 3 (GSK-3), a protein involved in energy metabolism, cell growth, and body pattern formation, which enhances wound healing via the *β*-catenin-dependent Wnt signaling pathway [46]. Both acute and chronic wounds include MMPs, which are categorized into eight families. They are essential for managing extracellular matrix deposition and breakdown, which is necessary for wound re-epithelialization.

The *Nephthea* sp. extract showed inhibitory activity towards GSK-3 compared with CHIR-99021 as a reference inhibitor, with IC_50_ values of 0.178 ± 0.009 µg/mL and 0.065 ± 0.003 µg/mL, respectively. Moreover, the enzymatic inhibition activity of the *Nephthea* sp. extract against MMP-1 opposing NNGH, with IC_50_ values of 0.258 ± 0.011 µg/mL and 0.055 ± 0.002 µg/mL, is illustrated in Figure 11.

### 2.5. Molecular Docking

As a first step to further determine the mode of action of the identified compounds as potential wound healing agents, a molecular-docking study was performed to determine the binding modes against glycogen synthase kinase and matrix metalloproteinase-1. The co-crystallized ligands *N*-(4-methoxybenzyl)-*N*′-(5-nitro-1,3-thiazol-2-yl) urea (TMU) for GSK-3 and methylamino-phenylalanyl-leucyl-hydroxamic acid (PLH) for MMP-1 proteins were reduced to ensure the validity of the docking parameters and methods to represent the position and orientation of the ligand detected in the crystal structure. The difference in the RMSD value between co-crystal ligands and the original co-crystal ligands was <2 Å, which approved the accuracy of the docking protocols and parameters, as illustrated in Table 3 and Figure 12, Figure 13, Figure 14 and Figure 15. All docking procedures and scoring were recorded according to our previous publications [47,48,49].

### 2.6. Physicochemical and ADME In Silico Study

Pharmacokinetic parameters should be incorporated with the significant filters, in order to further improve the chosen lead molecule transfer into drug candidates and permit lower failure rates in clinical trials because the majority of drug candidates fail during clinical development due to inadequate ADME features [17].

Based on docking results, the physiochemical properties and the drug likeness of the compounds chabrolohydroxybenzoquinone F (**13**) and isogosterone B (**9**) were studied. Moreover, ADME and toxicological studies were performed for these two promising compounds **9** and **13**. The physicochemical properties of chabrolohydroxybenzoquinone F (**13**) and isogosterone B (**9**) were generated based on website data and are presented in a Bioavailability Radar Chart in Figure 16. The results depict that both compounds have excellent physicochemical properties and fulfil most of the criteria documented in Table 4. In addition, both compounds **9** and **13** showed acceptable medicinal–chemistry properties, drug likeness (fitting with the Lipinski and Pfizer rules), and ADME properties (see Appendix A).

## 3. Materials and Methods

### 3.1. Soft Coral Material and Chemicals

In January 2020, during snorkeling activity off the Red Sea coasts in Hurghada, Egypt, the soft coral *Nephthea* sp. was collected. Dr. El-Sayed Abed El-Aziz graciously provided an authentication for the specimen (Department of Invertebrates Lab., National Institute of Oceanography and Fisheries, Red Sea Branch, Hurghada, Egypt). The Pharmacognosy Department of the Faculty of Pharmacy at Deraya University created, maintained, and recorded the voucher specimen under the number NS-19-1-2020. Until examination, it was kept at −10 °C. For marine extraction, analytical-grade solvents (methanol, methylene chloride) were used.

#### Extraction and Preparation

The frozen marine organism (500 g) was divided into smaller segments and was then repeatedly extracted using a methanol–methylene chloride (1:1) solution until it was completely exhausted. Finally, the solvent was evaporated using a rotary vacuum evaporator at a 60 °C water bath temperature, to yield 30 g of a concentrated extract.

### 3.2. Metabolomic Profiling Study

Metabolomic fingerprinting was performed on the *Nephthea* sp. extract using an Acquity Ultra Performance Liquid Chromatography system coupled to a Synapt G2 HDMS quadrupole time-of-flight hybrid mass spectrometer (Waters, Milford, MA, USA), as discussed earlier (Hassan et al., 2022) [16].

Chromatographic separation was carried out on a BEH C18 column (2.1 × 100 mm, 1.7 μm particle size; Waters, Milford, USA) with a guard column (2.1 × 5 mm, 1.7 μm particle size) and a linear binary solvent gradient of 0–100% eluent B over 6 min at a flow rate of 0.3 mL min^−1^, using 0.1% formic acid in water (*v/v*) as solvent A and acetonitrile as solvent B. The injection volume was 2 μL and the column temperature was 40 °C. To convert the raw data into separate positive and negative ionization files, MS Convert software was used. The files were then imported to the data mining software MZ mine 2.10 for peak picking, deconvolution, deisotoping, alignment, and formula prediction. The database used for compound identification was the Dictionary of Natural Products and Competitive Fragmentation Modeling for Metabolite Identification (CFM-ID) [18].

### 3.3. Isolation of the Major Metabolites

Silica gel 60 (63–200 m, E. Merck, Sigma Aldrich, Germany) was used for column chromatography (CC), while silica gel GF254 for thin-layer chromatography (TLC) (El-Nasr Company for Pharmaceuticals and Chemicals, Cairo, Egypt) was employed for vacuum liquid chromatography (VLC). Pre-coated silica gel 60 GF_254_ plates (E. Merck, Darmstadt, Germany; 20 × 20 cm, 0.25 mm in thickness) were used instead of thin-layer chromatography (TLC), which was driven out. Splashing with a *p*-anisaldehyde reagent (85:5:10:0.5, absolute ethanol: sulfuric acid: glacial acetic acid: *p*-anisaldehyde), followed by heating to 110 °C, was used to visualize spots [2]. For ^1^H and ^13^C NMR analyses, the following equipment was used: a Bruker AVIIIHD 400 FT-NMR spectrometer (400/3) from Japan. The measurements were made using CDCl_3_ as a deuterated solvent. The chemical shift values are represented in ppm (NMR Laboratory, Faculty of Pharmacy, Cairo University).

### 3.4. Pharmaceutical Preparation

#### 3.4.1. Preparation and Characterization of Soft Coral *Nephthea* sp. Loaded with Pectin Nanoparticles (SCN-LPN)

Pectin from citrus or apple (galacturonic acid > 75%) was purchased from Fischer Scientific, USA. Calcium chloride (CaCl_2_), disodium hydrogen phosphate, potassium dihydrogen phosphate, potassium chloride, and sodium chloride were purchased from El-Nasr Chemical Company (Cairo, Egypt). A dialysis tubing cellulose membrane (12,000–14,000 molecular weight cut-off) was purchased from Sigma Aldrich, Germany. Pluronic^®^F127 (PF127) and Pluronic^®^F68 (PF 68) were purchased from LOBA Chemie, India. Methanol absolute (99%) was purchased from the United Company for Chemical Medical Preparation (Cairo, Egypt).

##### Preparation of SCN-LPN

SCN-LPN was prepared with the ion-gelation technique with slight modification [20]. Briefly, 10 mL of a 0.5% *w/v* pectin solution was mixed with 3 mL of methanol (as the organic solvent), which contained 20 mg of the soft coral *Nephthea* sp. (SCN) extract. The above-described system was then treated drop-wise with an aqueous CaCl_2_ solution (as a cross-linker) in different ratios, while being continuously stirred on a magnetic stirrer (Wisd WiseStir Lab, Instruments, UDA) for 30 min at 50 °C for 1000 rpm, until complete evaporation of the methanol, then filtrated through a filter membrane with a pore size of 0.45 µm. Table 5 lists the composition of various SCN-LPN formulations.

#### 3.4.2. Characterization of SCN-LPN

##### Determination of % SCN Entrapment Efficiency (% E.E.)

The % E.E. of SCN within the freshly prepared SCN-LPN was estimated indirectly by calculating the supernatant amount of SCN [25]. An aliquot of 1 mL of the freshly prepared SCN-LPN was centrifugated at 4 °C for 17,000 rpm for 30 min with cooling (PORISPIN 17R, Novapro Co. Ltd., Bucheon-si, Republic of Korea). The supernatant was separated and filtrated through a filter membrane (0.45 µm). The SCN was measured with a UV-visible spectrophotometer (JASCO, V-730, Japan) against methanol as a blank at a maximum wavelength (λ_max_) of 415 nm. The % E.E. was calculated using the following equation:(1)$$\%\;E.E.=\frac{\mathrm{total}\;\mathrm{amount}-\mathrm{unentrapped}\;\mathrm{amount}}{\mathrm{total}\;\mathrm{amount}}\times 100$$

##### Determination of Particle Size (PS), Polydispersity Index (PDI), and Zeta Potential (ZP)

Prior to measuring the PS and PDI, the freshly prepared SCN-LPN formulations (1.0 mL) were appropriately diluted with deionized water and vortexed to have a suitable scattering intensity. The ZP of the prepared SCN-LPN formulations was determined by observing their electrophoretic mobility in an electrical field, using a Malvern Zetasizer Nano (Nano-ZS, Malvern Instruments Ltd., Malvern, UK) at an angle of 90° in a 10 mm diameter cell at 25 °C [22].

##### Transmission Electron Microscopy (TEM)

Transmission electron microscopy (TEM, JEOL, JEM-1011, and Tokyo, Japan) was used to evaluate the size and the distribution of the prepared samples.

##### Fourier Transform Infrared Spectroscopy (FTIR)

The chemical nature of the prepared samples was recorded for the SCN extract, pectin, CaCl_2_, and selected lyophilized SCN-LPN formulation and unloaded formulation using a Fourier transform infrared (FTIR) spectrometer (VERTEX 80v, Bruker, Billerica, MA, USA).

#### 3.4.3. Preparation and Characterization of Soft Coral *Nephthea* sp. Loaded with Pectin Nanoparticles *In-Situ* Gel (SCN-LPN-ISG)

##### Preparation of SCN-LPN-ISG

The cold method was used to prepare SCN-LPN-ISG with certain modifications [50]. In brief, 25% *w/v* of thermosensitive gel composed of a mixture of Pluronic^®^ F127 (PF127) and Pluronic^®^F68 (PF68) in a suitable ratio was sprinkled over the selected SCN-LPN formulation, surrounded by an iced beaker at 4 °C and stirred at 400 rpm on a magnetic stirrer until a homogeneous system without lumps was obtained. The SCN-LPN-ISG thus obtained was stored overnight in a refrigerator for further studies.

#### 3.4.4. Characterization of SCN-LPN-ISG

##### Visual Characterization, pH, and Drug Content (%) Analysis

The prepared SCN-LPN-ISG was visually checked for clarity, homogeneity, and color against a black and white background [51]. For pH measurement, a pH meter (Schott CG 840, Mainz, Germany) was used by dipping an electrode inside the prepared SCN-LPN-ISG. For the drug content analysis, 1 g from freshly prepared SCN-LPN-ISG was taken and suspended in 30 mL of PBS (pH = 7.4) with vigorous shaking for 12 h. The amount of SCN was measured spectrophotometrically against PBS (pH = 7.4) as a blank at λ_max_ = 415 nm, using a UV-Vis spectrophotometer. The drug content (%) was calculated using the following equation:(2)$$\mathrm{Drug}\;\mathrm{content}\;\left(\%\right)=\frac{\mathrm{actual}\;\mathrm{amount}\;\mathrm{of}\;\mathrm{SCN}}{\mathrm{theortical}\;\mathrm{amount}\;\mathrm{of}\;\mathrm{SCN}}\times 100$$

##### *Sol–Gel* Transformation Temperature

The inversion technique was used to detect the *sol–gel* transformation temperature [52]. The freshly prepared SCN-LPN-ISG (1.0 mL) was put into a sealed test tube. It was immersed in thermostatic water, while gradually raising the temperature from 25 °C to 37 °C, the temperature at which there was no movement of the liquid when the test tube was tilted up perpendicularly as recorded [53].

##### Rheology Properties

A cone and plate Brookfield viscometer (Model DV-III Rheometer, spindle CPE-40, Middleboro, MA, USA) with spindle (40) was used to estimate the viscosity of the freshly prepared SCN-LPN-ISG. It was measured at different temperatures (such as 25 °C and 37 °C), while varying the angular velocity between 1 rpm and 10 rpm. This was conducted to compare the viscosity before and after gelling formation [54].

### 3.5. In Vitro Release Studies

The percentage of the cumulatively released SCN amount during different time intervals from the prepared selected SCN-LPN and SCN-LPN-ISG formulations against SCN extract suspension was studied, using the dialysis tube technique. In brief, a specific amount from the prepared formulations was dispersed in 2 mL of PBS (pH = 7.4) and placed in a dialysis bag made of a cellophane membrane (12,000–14,000 molecular weight cut-off), which was blocked well at both ends. The dialysis bag was immersed in a beaker containing 500 mL of PBS (pH = 7.4), and stirred at a constant rate of 100 rpm on a magnetic stirrer at 37 °C for 8 h. An aliquot of the sample (3 mL) was withdrawn at specific time intervals (0.5, 1, 2, 3, 4, 5, 6, and 8 h) and replaced with fresh PBS (pH = 7.4). The amount of SCN released was measured spectrophotometrically against PBS (pH = 7.4) as a blank at λ_max_ = 415 nm, using a UV-visible spectrophotometer.

### 3.6. In Vivo Diabetic Wound Healing Study

#### 3.6.1. Experimental Animals and Ethical Statement

In this study, 24 male white Wistar rats weighing 100–120 g were utilized. They were purchased from the animal house of the Faculty of Science, Cairo University. The animals were housed separately, in steel mesh cages; kept under standard conditions—ventilation, temperature (25 ± 2 °C), humidity (60–70%), and light/dark conditions (12/12 h); and fed with a standard rat fresh diet along with clean drinking water. They were acclimatized for a period of 10 d before the beginning of this study. The rats were randomly divided into four groups (six rats each).

All surgical procedures and handling were approved by the research ethics committee of the Faculty of Pharmacy, Cairo University (Cairo, Egypt; under the number MP-3225) in compliance with the *Guide for the Care and Use of Laboratory Animals* published by the US National Institutes of Health, 8th edition (NIH Publication National Research, 2011). All efforts were made to minimize animal pain, discomfort, and suffering.

#### 3.6.2. Diabetic Model Induced

To induce type 2 diabetes in experimental rats, streptozotocin (STZ) was injected simultaneously with normal chow and a high-fat diet (60% calories as fat, 58Y1, Test Diet) for 2 weeks. A single intraperitoneal injection of 60 mg/kg STZ was given with a recovery high-fat diet during the experiment. The confirmation of the occurring type 2 diabetes in rats was ensured by measuring the blood glucose level with a glucometer (Accu-Chek Active, Berlin, Germany) 1 week after STZ injection [55].

#### 3.6.3. Circular Excision Wound Model

The back hair of all rats was shaved off. All rats were anesthetized with ether using inhalation before a full-thickness skin wound was made in the dorsum (in the subscapular area) using a 1 cm diameter punch. The dorsum position was selected to make the wound to ensure that they were inaccessible by the nails and the mouth, which prevented self-licking [56]. All rats were topically treated once daily day-after-day for 2 weeks.
**The first group** was considered as a control (untreated).**In the second group**, the rats received a prepared extract of *Nephthea* sp. (applied as small pieces of sterile gauze macerated in 1.5 gm of the extract) and two pieces (30 mg) per wound dressing day-after-day for 15 d.**The third group** received 1 cm from SCN-LPN-ISG (Conc.: 2 mg/mL), changed day-after-day for 15 d.**The fourth group** served as a positive control, receiving MEBO ointment (as the market reference drug, 100 mg/wound, daily, Gulf Pharmaceutical Industries Company, Ras Al Khaimah, United Arab Emirates) day-after-day throughout the experiment period.

The blood samples were collected on days 7 and 15 and sera samples were isolated with centrifugation at 5000 rpm for 10 min and kept at 80 °C. The animals were euthanized, and skin tissues were taken at the end of the experiment.

#### 3.6.4. Wound Concentration Analysis

On days 0, 3, 6, 9, 12, and 15 of the experiment, each wound was photographed using a Sony DSC-WX70/P digital camera (Berlin, Germany) and a scale bar was used to measure the contraction of the injured area compared with the situation on day 0. By measuring the injured areas with specialized software for this morphometric study, the percentage of wound contraction for each rat was estimated using the formula below [57].
Wound concentration (%) = [(initial wound area − analyzed area)/initial wound area] × 100

### 3.7. Histological and Histochemical Assessment Studies

Skin tissue samples (from wound sites) were taken from all rats in different groups and fixed in 10% formalin saline for 24 h. Washing was performed in tap water; then, they were subjected to dissentingly serial dilutions of alcohol (methanol, ethanol, and absolute ethanol) that were used for dehydration. Specimens were cleared in xylene and embedded in paraffin at 56 °C in a hot-air oven for 24 h. Paraffin beeswax tissue blocks were prepared for sectioning at a 4–5 μm thickness with a rotary LEITZ microtome. The obtained tissue sections were collected on glass slides, deparaffinized, and stained with hematoxylin and eosin for examination through the light microscope [58].

### 3.8. Inflammatory Biomarker Assays

The levels of the nuclear factor kappa-light-chain-enhancer of activated B cells (NF-κB), interleukin 6 (IL-6), tumor necrosis factor-α (TNF-α), and interleukin-1β (IL-1β) in liver tissues were analyzed with ELISA kits obtained from Cusabio (Wuhan, China), as per the official manufacturer’s method [59].

### 3.9. Inhibition Activity of Nephthea sp. Extract against GSK-3 and MMP-1 Enzymes (In Vitro)

To assess the inhibition activity of the *Nephthea* sp. extract against glycogen synthase kinase (GSK-3) and matrix metalloproteinase-1 (MMP-1) enzymes, CHIR-99021 and NNGH were used as positive reference inhibitors, respectively. The assay was carried out with kits of Kinase-Glo^®^ (Promega Corporation, Madison, WI, USA) according to the procedure reported in the literature [60].

### 3.10. In Silico Studies

#### Molecular Docking and Pharmacokinetics, “ADME” Activity

The structures of all tested compounds were modeled using Chem Sketch software (http://www.acdlabs.com/resources/freewar (accessed on 19 January 2023)). The structures were optimized and energy minimized using VEGAZZ software [61]. The optimized compounds were used to perform molecular docking to elucidate the potential activity against glycogen synthase kinase (GSK-3) and matrix metalloproteinase-1 (MMP1), to analyze the proposed wound healing activity for the most promising compounds by targeting the enzyme. The three-dimensional structure of the molecular target was obtained from the Protein Data Bank (PDB) (www.rcsb.org (accessed on 19 January 2023)): (PDB: 1Q5K, https://www.rcsb.org/structure/1Q5K (accessed on 19 January, 2023)), (PDB: https://www.rcsb.org/structure/1HFC (accessed on 19 January 2023)). The steps for receptor preparation included the removal of heteroatoms (water and ions), the addition of polar hydrogen, and the assignment of a charge. The active sites were defined using grid boxes of appropriate sizes around the bound co-crystal ligands. The docking study was performed using AutoDock Vina [62], and Chimera for visualization [63]. All docking procedures and scoring were recorded according to our previous publications [64].

To identify the biological targets for the most promising tested compounds, we employed searching in a database function integrated into Swiss Institute Bioinformatics tools [65]. Further, absorption, distribution, metabolism, and excretion “ADME” were hypothetically calculated [66].

### 3.11. Statistical Analysis

All experiments were conducted in triplicate, and the data are expressed as the mean ± SD. The one-way analysis of variance (ANOVA) was followed by the Duncan test to calculate statistical differences between the different treatments. *p* < 0.05 means that there was a significant difference between the data. GraphPad Prism 8.0 (GraphPad Prism Software Inc., San Diego, CA, USA) was used to visualize the results. Statistical differences in the wound healing area at *p* < 0.01 were observed between the treatment and control groups.

## 4. Conclusions

Herein for the first time, the preparation and characterization of a polymeric (pectin) nanoparticle-loaded marine soft coral *Nephthea* sp. extract is reported. An *in situ* gel formula was formed to determine its potential stimulation activity toward diabetic wound healing. The UPLC-Q/TOF-MS fingerprint of the extract represents the secondary metabolite profile of the soft coral. This is the first documentation of the isolation of three fatty acids and one phthalate compound from *Nephthea* sp. *In silico* and ADME studies of the identified compounds evidenced their mechanism of action to be a wound healing enhancer. Our results indicate that *Nephthea* sp. is a rich source of effective metabolites, making it a promising candidate as a wound healing drug. We hope that this work will raise awareness of natural marine sources that can be used to speed up the healing of chronic diabetic wounds and encourage further clinical trials to evaluate the safety of *in situ* gel nanoparticles against synthetic medications to treat chronic wounds.

## Data Availability

Data is contained within this article and Appendix A.

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
