# Peer review of "Pectin Nanoparticle-Loaded Soft Coral *Nephthea* sp. Extract as *In Situ* Gel Enhances Chronic Wound Healing: *In Vitro*, *In Vivo*, and *In Silico* Studies"

_pharmaceuticals, 2023, doi:10.3390/ph16070957_

Round 1
Reviewer 1 Report
In the present study, the authors developed Nephthea sp. methanol-methylene chloride extract loaded pectin nanoparticles (LPNs) to promote wound healing. Extensive studies were conducted in-vitro, in vivo, and in-silico. The effective compounds in Nephthea sp. methanol-methylene chloride extract were isolated and identified, and the in vivo wound-healing effects were proved on animal studies. The study provided extensive evidence supporting the therapeutic potential of LPNs, which is suitable for publication in present form.
Author Response
|
We are grateful positive decision toward our work and very happy to satisfy your respective opinion. |

Reviewer 2 Report
Authors presented the wound healing results with the title of “Pectin Nanoparticles Loaded Soft Coral Nephthea sp. Extract as In-situ Gel Enhances Chronic Wound Healing: In-vitro, In- vivo, and In-silico Studies”
1. Figure 1, Size distribution and zeta potential, the observed data for three peaks with different values, but the image has only one peak. Authors advised to double-check the results.
2. Metabolomic study, number of shorts, wavelength, and source of ionization to be explained in more detail for clear understanding.
3. Authors have mentioned and listed the molecules (Table 1) based on their molecular weight. If authors used positive mode UPLC-Q/TOF-MS, each molecule should be ionized. Authors advised presenting the molecular weight along with ions in the table.
4. Authors advised to present a few bare molecules(commercial) results of UPLC-Q/TOF-MS as a comparative results to confirm the extracted molecules.
5. Authors have used 1.5gm of extract to direct wound healing. Question is. Several factors may influence the concentration of each molecule among crude. How do authors confirm each molecule’s concentration?
6. From the wound healing results compared with MEBO cream, Which parameter is overcome by the existing burn cream?
7. Figure 10. Authors evaluated the inflammatory protein level compared with MEBO cream-treated group. It’s surprising that protein levels exist after 15 days in the MEBO group, but the digital image is clearer after 15 days of treatment. Could authors explain clear comparative results of Figure 10 with Figure 7?
Reviewer 3 Report
In this manuscript, the authors mainly used ionic gelation technique and created Nephthea sp. methanol-methylene chloride extract loaded pectin nanoparticles (LPNs). Through wound healing experiments on diabetic mice, the shed light for the first time on the in-vivo diabetic wound healing potential activity of natural marine soft coral polymeric nanoparticles in-situ gel using an excision wound model. The results indicate that Nephthea sp. is a rich source of effective metabolites, making it a promising candidate as a wound healing drug. The novelty of the study is to investigate the chronic wound healing stimulation activity of pharmaceutical preparation of polymer nanoparticles loaded Nephthea sp. which was obtained from the Red Sea region. There are still some problems with the experimental methods, data processing and references in the manuscript. This manuscript could be accepted after Major revision. There still have some other questions were shown below:
1. The abstract section shows too much details and data, ignoring the research background, research methods and methodology.
2. Nearly half of the references are not published within the last five years, to ensure the novelty of the references. The pectin reference should be referred this (Food Hydrocolloids,143(2023): 108901.).
3. Page 2, line 70 mentions that pectin is derived from the cell walls of some citrus plants, which is not rigorous. Please refer this reference (Food Hydrocolloids,133(2022): 107910).
4. Page 2, line 89 refers to the preparation of extracts using the ionic gel method, and a brief explanation of the ionic gel method should be provided.
5. Page 3, Scheme 1 is not concise enough for the overview of each experiment, while the graphical content does not show the data clearly.
6. Page 4, Figure 1. does not clearly show the data mentioned in line 143
7. Page 4, 2.1.1, zone electron diffraction (SAED) was selected to verify the amorphous nature of the prepared formulations, but the specific method and details about the verification are not reflected in the later text.
8. Page 5, Figure 3. should be marked with the data about the absorption peaks at 415nm and 350nm.
9. Page 6, line 187. should be PF®127 and PF®68 for further explanation.
10. Page 8, Figure 6. The chemical structure formula is not well arranged, and it would be easier to understand if the molecular formula were labeled.
11. Page11, 2.3.1. The experiment on wound healing lacks detailed data, such as the dosage of SCN-LPN-ISG, making it difficult to replicate the experiment.
12. Page 14, Figure 10. Experiments on the inflammatory response to the wound healing process should be followed by a control group, such as the data from day11 in day7-day15, to better reflect the effect of the preparation.
13. “2.5. Molecular Docking”. It should refer this reference when analyze the interaction information (Food Bioscience. 43(2021), 101313.).
14. Page17, Table 4. The data in Table 4. is not clear enough to facilitate the reader's access to information.
15. Page21. The table number should be corrected.
16. Page 26, line 691, lack of validation for the "ADME" assumption calculation, and details of the assumption calculation.
In this manuscript, the authors mainly used ionic gelation technique and created Nephthea sp. methanol-methylene chloride extract loaded pectin nanoparticles (LPNs). Through wound healing experiments on diabetic mice, the shed light for the first time on the in-vivo diabetic wound healing potential activity of natural marine soft coral polymeric nanoparticles in-situ gel using an excision wound model. The results indicate that Nephthea sp. is a rich source of effective metabolites, making it a promising candidate as a wound healing drug. The novelty of the study is to investigate the chronic wound healing stimulation activity of pharmaceutical preparation of polymer nanoparticles loaded Nephthea sp. which was obtained from the Red Sea region. There are still some problems with the experimental methods, data processing and references in the manuscript. This manuscript could be accepted after Major revision. There still have some other questions were shown below:
1. The abstract section shows too much details and data, ignoring the research background, research methods and methodology.
2. Nearly half of the references are not published within the last five years, to ensure the novelty of the references. The pectin reference should be referred this (Food Hydrocolloids,143(2023): 108901.).
3. Page 2, line 70 mentions that pectin is derived from the cell walls of some citrus plants, which is not rigorous. Please refer this reference (Food Hydrocolloids,133(2022): 107910).
4. Page 2, line 89 refers to the preparation of extracts using the ionic gel method, and a brief explanation of the ionic gel method should be provided.
5. Page 3, Scheme 1 is not concise enough for the overview of each experiment, while the graphical content does not show the data clearly.
6. Page 4, Figure 1. does not clearly show the data mentioned in line 143
7. Page 4, 2.1.1, zone electron diffraction (SAED) was selected to verify the amorphous nature of the prepared formulations, but the specific method and details about the verification are not reflected in the later text.
8. Page 5, Figure 3. should be marked with the data about the absorption peaks at 415nm and 350nm.
9. Page 6, line 187. should be PF®127 and PF®68 for further explanation.
10. Page 8, Figure 6. The chemical structure formula is not well arranged, and it would be easier to understand if the molecular formula were labeled.
11. Page11, 2.3.1. The experiment on wound healing lacks detailed data, such as the dosage of SCN-LPN-ISG, making it difficult to replicate the experiment.
12. Page 14, Figure 10. Experiments on the inflammatory response to the wound healing process should be followed by a control group, such as the data from day11 in day7-day15, to better reflect the effect of the preparation.
13. “2.5. Molecular Docking”. It should refer this reference when analyze the interaction information (Food Bioscience. 43(2021), 101313.).
14. Page17, Table 4. The data in Table 4. is not clear enough to facilitate the reader's access to information.
15. Page21. The table number should be corrected.
16. Page 26, line 691, lack of validation for the "ADME" assumption calculation, and details of the assumption calculation.
Author Response
|
We thank our reviewer for his positive opinion of our work, we addressed all comments and hope to your satisfaction, Please see the attachment |

Round 2
Reviewer 3 Report
The author has responded to the reviewer's comments point by point. It can be accepted in the current revision.
The author has responded to the reviewer's comments point by point. It can be accepted in the current revision.